# Bio-Fermented Malic Acid Facilitates the Production of High-Quality Chicken via Enhancing Muscle Antioxidant Capacity of Broilers

**DOI:** 10.3390/antiox11122309

**Published:** 2022-11-22

**Authors:** Kai Qiu, Weizhen He, Haijun Zhang, Jing Wang, Guanghai Qi, Naiwei Guo, Xin Zhang, Shugeng Wu

**Affiliations:** 1National Engineering Research Center of Biological Feed, Institute of Feed Research, Chinese Academy of Agricultural Sciences, Beijing 100081, China; 2Technology Department, Anhui Sealong Biotechnology Co., Ltd., Bengbu 233080, China; 3State Key Laboratory of Animal Nutrition, College of Animal Science and Technology, China Agricultural University, Beijing 100193, China

**Keywords:** malic acid, acidifier, antioxidant capacity, water-holding capacity, meat quality, broilers

## Abstract

Malic acid, an intermediate of the tricarboxylic acid (TCA) cycle, is a promising acidifier with strong antioxidant capacity. This study aimed to evaluate the effects of bio-fermented malic acid (BFMA) on promoting the body health, performance and meat quality of broilers. A total of 288 one-day-old Arbor Acres male broiler chicks were randomly divided into four treatments with six replicates in each. Every replicate had 12 chicks. Four experimental diets contained 0, 4, 8, and 12 g/kg BFMA, respectively. During the 42-day trial, mortality was recorded daily, feed intake and body weight of each replicate being recorded every week. Blood samples were collected on days 21 and 42 for chemical analysis. After slaughter at the age of 42 days, the carcass traits and meat quality of the broilers were measured, breast muscle samples were collected for the determination of antioxidant capacity, and cecal digesta were pretreated for microbiota analysis. Dietary BFMA significantly increased feed intake and daily gain, and decreased feed conversion ratio and death and culling ratio of the broilers at the earlier stage. The water-holding capacity of breast muscle indicated by the indexes of dripping loss and cooking loss was significantly increased by BFMA, especially at the addition level of 8 g/kg. Dietary BFMA significantly decreased the activity of superoxide dismutase and contents of immunoglobulin A and glutathione, and increased contents of immunoglobulin G and M in serum of the broilers. The contents of glutathione, inosinic acid, and total antioxidant capacity and the activities of glutathione-Px and superoxide dismutase were significantly increased by dietary BFMA, with the level of 8 g/kg best. The diversity of cecal microbiota of broilers was obviously altered by BFMA. In conclusion, as one of several acidifiers, addition of BFMA in diets could improve the performance and body health of broilers, probably by reinforcing immunity and perfecting cecal microbiota structure. As one of the intermediates of the TCA cycle, BFMA increases the water-holding capacity of breast muscle of broilers, probably through reducing lactate accumulates and enhancing antioxidant capacity.

## 1. Introduction

Over the past few decades, a variety of additives with antimicrobial and/or growth-promoting effects have been widely used in poultry feed during commercial grow-out to enhance the economic benefits [1]. Pathogens develop resistance to antibiotics at a rate much faster than the discovery of new antibiotics, and resistant microorganisms can spread between food-producing animals and humans [2]. Antibiotic resistance and residues in broiler production have been taken as one of the biggest challenges of human health [3,4,5]. As the voice of antibiotic-free food is gradually increasing, it is a global trend in animal production to reduce or ban the use of antibiotics for growth in feed [6,7,8,9]. Nowadays, researchers have focused on identifying functional feed additives as alternatives to synthetic antibiotics in broiler production, such as probiotics, prebiotics, synbiotic, organic acid, bioactive peptide, emulsifiers, enzymes, and some bioactive substances [10].

The addition of acidifier in the diet of broilers could minimize inflammation-induced damage, prevent mortalities from excessive intestinal lesions and oxidative stress, and improve growth performance [11,12]. Antibiotics promote abdominal fat accumulation in broilers [13], while dietary supplementation with acidifiers reduced abdominal fat mass deposition and improved the meat quality of broiler chickens besides growth performance and intestinal health [11,14]. Acidified drinking water can compensate for gastric acidity, control pathogenic bacteria, and improve production parameters in broilers [15,16]. The acidifier, benzoic acid, was found as a substitute for antibiotics to improve growth performance, antioxidation, nutrient digestion and gut microbiota of yellow-feathered broilers [17]. Humic acid was demonstrated to be valuable in improving the production and health safeguarding of livestock and poultry [18]. Organic acid mixture could control *Salmonella typhimurium* infection and maintain efficient growth performance of broilers [19]. Contaminated poultry meat is considered to be the main source of human infection with *Campylobacter* spp. The acidification of drinking water through the addition of organic acids has been reported to decrease the risk of *Campylobacter* spp. colonization in broiler flocks [20,21]. In-feed inclusion of medium chain fatty acids, together with vaccination of a live attenuated coccidiosis vaccine, reduced necrotic enteritis in broilers [22]. A microencapsulated blend of organic acids and essential oils could improve the quality and shelf life of poultry meat [23]. Therefore, acidifiers are promising alternatives to antibiotics for growth performance, antioxidant status, and intestinal health of broilers.

As an intermediate of the tricarboxylic acid (TCA) cycle, malic acid is one of the most promising and renewable chemicals. L-malic acid is an important component of a vast array of food additives, antioxidants, disincrustants, pharmaceuticals, and cosmetics. Owing to the abundance of malic acid, tomato seed flour was used as a value-added food ingredient for its radical scavenging, anti-inflammatory and gut microbiota modulating properties [24]. Dietary supplementation of malic acid showed no significant effect on the growth of broilers [25]. However, the effects of antioxidant capacity of malic acid on the cecal microbial community and meat quality of broilers remain largely unknown.

## 2. Materials and Methods

### 2.1. Animals and Experimental Design

The bio-fermented malic acid (BFMA) was produced in Anhui Xuelang Biotechnology Co., Ltd. (Bangbu, China) by the fermentation of *aureobasidium pullulans* CCTCC NO:M 2,012,223 with the substrate containing glucose, citric acid, and corn starch. The BFMA was an unrefined malic acid product designed for the feed additive, the middle product during the production of pure malic acid from the fermentation products. The BFMA contained 20% L-malic acid and 80% carrier (1% bacterial protein, 2% polysaccharide, 2% fermentation residue, and 75% zeolite powder) [26]. The carrier was used in the present study to balance the dose of BFMA among experimental groups. A total of 288 one-day-old Arbor Acres (AA) male broiler chicks were randomly divided into 4 treatments; each treatment had 6 replicates. Each replicate had 12 chicks, raised in a wire floor cage (110 × 100 × 55 cm). The control group was fed a corn–soybean meal basal diet containing 12 g/kg carrier of malic acid, and the other three treatment groups were fed the basal diets with the carrier being substituted with 4, 8, and 12 g/kg BFMA, respectively. The feeding trial lasted for 42 days, and was divided into 3 stages: days 0–14 (starter period), days 15–28 (grower period), and days 29–42 (finisher period). Accordingly, three basal diets were formulated to meet the nutrient requirements of AMINOChick^®^2.0 and Chinese Feeding Standard of Chicken (NY/T, 33-2018). The feed was made into granules using a cold pelleting process before being given to the broilers. Ingredient and calculated nutrient compositions of basal diets are listed in Table 1.

### 2.2. Birds Management, Data Record, and Sample Collection

The feeding trial was carried out at the Nankou Test Base of the Chinese Academy of Agricultural Sciences (Beijing, China). The chicks were housed in three-layer cages and had free access to feed and water through individual feeders and drinkers in each cage. The experimental house was kept in 23 h of light and 1 h of darkness every day. The temperature was controlled at 34–35 °C in the first week, then decreased by 2 °C every week, and finally controlled between 24–26 °C. The relative humidity was 45–55%. According to the routine immunization program and feeding management, the health status of the birds was observed daily. Chicks were vaccinated with inactivated Newcastle disease vaccine on days 7 and 21, and inactivated infectious bursal disease vaccine on days 14 and 28. During the trial, the mortality of birds and their body weight were recorded daily. Feed intake (FI) and BW of each replicate were recorded every week. Average daily feed intake (ADFI), average daily gain (ADG), feed conversion ratio (FCR), and death and culling ratio (DCR) were calculated according to the record.

At day 21 and 42 of the trial, the birds were weighed after 8 h fasting. One bird was randomly selected from each replicate, and 10 mL of nonanticoagulated blood was collected through the wing vein, placed at an incline for 30 min at room temperature, centrifuged at 3000 rpm for 15 min to obtain serum, and stored at −20 °C for use in the determination of biochemical indexes. On day 42, the broilers from which blood samples had been collected were then slaughtered for the measurement of carcass traits and meat quality. The cecal digesta were collected and stored at −80 °C for microbiological analysis.

### 2.3. Carcass Traits and Meat Quality

After being slaughtered at the age of 42 days, the carcass, evisceration, breast muscle, leg muscle, and abdominal fat of broilers were weighed, and the carcass rate, evisceration rate, complete evisceration rate, chest muscle rate, leg muscle rate and abdominal fat rate were calculated. The formulas are as follows:

Dressing = (carcass weight/live weight) × 100%

Evisceration rate = (eviscerated weight/live weight) × 100%

Breast muscle rate = (pectoral muscle weight/evisceration weight) × 100%

Leg muscle rate = (leg muscle weight/evisceration weight) × 100%

Abdominal fat rate = (abdominal fat weight/evisceration weight) × 100%

The right pectoralis major muscle of each broiler was taken after slaughter, and the color (*L**, brightness; *a**, Redness; *b**, Yellowness) was measured immediately with a fully automatic colorimeter (CR-410, Konica Minolta, Tokyo, Japan) with a whiteboard for reference. Using a portable pH meter, the pH probe (pH-star, DK2730, Herlev, Denmark) was inserted into the meat for measurement; each piece of meat was measured at 3 different positions, and the average value was taken as the final result, which was recorded as pH_45 min_. Then, the meat was placed in a refrigerator at 4 °C for 24 h; subsequently it was taken out, each piece of meat was measured at 3 different locations, and the average value was taken as the final result, which was recorded as pH_24 h_. After slaughter, the middle part of the left pectoralis major muscle was taken, trimmed into a length × width × thickness of 30 mm × 15 mm × 5 mm, and weighed (W1). Then one end of the meat sample was caught with an iron wire so that the muscle fibers were vertically upward. This was placed into an air-filled plastic bag with no contact with the bag wall, and the bag mouth was tied. The meat sample was hung in a refrigerator at 4 °C for 24 h, then taken out and dried with filter paper, the meat was weighed (W2), and its dripping water loss was calculated as (W1 − W2)/W1.

### 2.4. Immune and Antioxidant Indicators of Serum and Breast Muscle

Serum antioxidant indicators include glutathione peroxidase (GSH-Px), glutathione (GSH), superoxide dismutase (SOD), malondialdehyde (MDA) and total antioxidant capacity (T-AOC). Serum immune capacity indicators include immunoglobulin A (IgA), immunoglobulin M (IgM) and immunoglobulin G (IgG). Breast antioxidant indicators include GSH-Px, GSH, SOD, MDA, T-AOC, and lactic acid. The assay was performed with the colorimetric method using an automatic biochemical analyzer (Model 7160, Hitachi Group, Tokyo, Japan), following the instructions of kits (Nanjing Jiancheng Bioengineering Institute, Nanjing, China). Information regarding the commercial kits is listed in Appendix A.

### 2.5. Determination of Inosinic Acid in Chicken

Inosinic acid (inosine monophosphate, IMP) is the major umami compound in chicken, having an important function in meat flavor formation. An Agilent TC-C18 chromatographic column (5 μm, φ4.6 mm × 250 mm) was used in the detection of high performance liquid chromatography (HPLC). The mobile phase was 0.05 mol/L pH 6.5 sodium dihydrogen phosphate buffer solution. The flow rate was 1 mL/min, and the column temperature was 30 °C. The UV detection wavelength was 254 nm.

### 2.6. DNA Isolation, PCR Amplification, and Miseq Sequencing

The total genomic DNA of cecal digesta was extracted using the Qiagen QIAamp DNA Kit in accordance with the manufacturer’s guidelines. The concentration of DNA samples was measured using a Nano-Drop 3000 spectrophotometer. The 16S rRNA gene was amplificated by polymerase chain reaction (PCR) using specific primers for the V3 region according to our previous report [27]. All PCR products were isolated using 2% agarose gels, and purified with a DNA gel extraction kit (Axygen, Hangzhou, China). Before sequencing, PCR products were mixed with the same proportion based on their DNA concentration determined by a QuantiFluor™-ST fluorescent quantitative system (Promega, Madison, WI, USA). The construction of PE amplicon libraries and sequencing were conducted on the Illumina Miseq platform (Majorbio Bio-Pharm Technology Co., Ltd., Shanghai, China).

### 2.7. Sequence Analysis

Before analysis, sequences were filtered based on their quality using QIIME (v1.9.1) with the following criteria: (1) the 300 bp reads truncated at any site with an average quality score <20 over a 50 bp sliding window, (2) exact barcode matching with no more than one nucleotide mismatch in primer matching, (3) only sequences overlapped longer than 10 bp assembled as to their overlap sequence. Sets of sequences with at least 97% identified were defined as operational taxonomic units (OTUs), and chimeric sequences were identified and removed using UCHIME. The taxonomy of each 16S rRNA gene sequence was performed using RDP Classifier (http://rdp.cme.msu.edu/ (accessed on 7 April 2022)) against the SILVA ribosomal RNA gene database with 70% as the confidence threshold.

### 2.8. Ecological and Statistical Analyses

The abundance of communities and sequencing data of each sample were assessed based on rarefaction curves. Alpha-diversity was analyzed using Mothur software based on the number of actually measured OTUs (Sobs), including community diversity (Shannon, Simpson), community richness (Chao, Ace), and sequencing depth (Good’s coverage). Analyses of beta diversity were performed, including partial least squares discriminant analysis (PLS-DA), principal coordinate analysis (PCoA), and nonmetric multidimensional scaling (NMDS). Differences between populations at phylum and genus levels were analyzed using a one-way ANOVA. *p* < 0.05 was considered statistically significant.

### 2.9. Statistical Analysis

The experimental data were analyzed using the one-way ANOVA program of SPSS Version 19.0 (SPSS, Chicago, IL, USA) and multiple comparisons were conducted using Tukey’s Test. Means were considered significantly different at *p* < 0.05, and a trend of change at 0.05 ≤ *p* < 0.10.

## 3. Results

### 3.1. Growth Performance

The performance of the broilers is shown in Table 2. At the beginning of the trial, the one-day old broiler chicks had similar BW between groups (*p* = 1.00). Dietary addition of BFMA significantly improved the BW of broilers on day 21 (*p* < 0.001), while it did not influence BW at day 42 (*p* > 0.05). During days 1 to 21, dietary supplementation of BFMA significantly increased ADFI and ADG, and decreased FCR and DCR of broilers as compared with the control group (*p* < 0.01). During days 22 to 42, the ADG of broilers fed diets supplemented with BFMA showed a tendency to decrease relative to the control group (*p* = 0.077), while the ADFI, FCR, and DCR was not influenced (*p* > 0.05). Over the whole trial period, dietary supplementation of BFMA showed no effects on the ADFI, ADG, FCR, and DCR of broilers (*p* > 0.05).

### 3.2. Carcass Traits and Meat Quality

The slaughter performance and meat quality of the broilers are shown in Table 3. Dietary supplementation of BFMA had no effects on live BW, carcass weight, dressing, eviscerated weight, eviscerated ratio, breast ratio, thigh ratio, or abdominal fat ratio of broilers (*p* > 0.05). The addition of BFMA in diets had no effects on the lightness or redness of breast muscle (*p* > 0.05), but had a significant effect on its yellowness (*p* < 0.05). Specifically, compared with the control group, the addition of 4 g/kg BFMA in the diet significantly increased the yellowness of breast meat (*p* < 0.05), but with increasing addition levels, the yellowness gradually decreased to the level of the control group. Dietary supplementation of BFMA had no effects on the pH value of breast muscle at 45 min, but increased it at 24 h after slaughter at the age of 42 days (*p* < 0.05). The dripping loss of the broilers’ breast muscle was decreased by dietary supplementation with BFMA (*p* < 0.05). Compared with the control group, the breast muscle of broilers fed the diet supplemented with 8 g/kg BFMA showed significantly reduced cooking loss (*p* < 0.05), but the breast muscle of those with 4 or 12 g/kg supplementation levels was not changed (*p* > 0.05).

### 3.3. Serum Antioxidant and Immune Capacity

The serum antioxidant indexes of the broilers are shown in Table 4. On days 21 and 42, dietary supplementation of BFMA had no effects on the activity of GSH-Px and the concentrations of MDA and T-AOC in serum (*p* > 0.05). On day 21, the content of GSH in serum of broilers showed a decrease with the supplementation of BFMA in diets (*p* = 0.096). The activity of SOD in the serum of broilers fed the diet supplemented with 12 g/kg BFMA was decreased compared with other groups (*p* < 0.05). However, on day 42, the content of GSH and the activity of SOD in serum were not influenced by the dietary supplementation of BFMA (*p* > 0.05).

The content of immunoglobulin in the serum of the broilers is shown in Table 5. On day 21, compared with the control group, dietary supplementation of 12 g/kg BFMA significantly decreased the IgA content in serum of broilers (*p* < 0.05), increased the IgG content (*p* < 0.05), and had a tendency of increasing IgM content (*p* = 0.075). On day 42, dietary supplementation of 8 or 12 g/kg BFMA significantly increased IgA and IgG levels in serum of broilers compared with the control groups (*p* < 0.05), but had no effects on the IgM level (*p* > 0.05).

### 3.4. Antioxidant Capacity of Breast Muscle

The contents of GSH, MDA, T-AOC, IMP, and lactic acid and the activities of GSH-Px and SOD in the breast muscle of the broilers are presented in Table 6. Dietary supplementation with BFMA significantly affected the indexes of breast muscle including GSH, GSH-Px, SOD, T-AOC, IMP, and lactic acid (*p* < 0.05), while it showed no effect on the content of MDA (*p* > 0.05). The specific differences are shown below. The addition of 8 g/kg BFMA in diet significantly increased the content of GSH in the breast muscle, while 4 or 12 g/kg BFMA had no effect relative to the control group. The GSH-Px activity in the breasts of broilers fed diets supplemented with 4, 8, or 12 g/kg BFMA was significantly enhanced compared with the control group. The broilers fed diets supplemented with 8 or 12 g/kg BFMA showed significantly increased SOD activity and IMP content in the breast muscle, while the group fed 4 g/kg BFMA showed no difference with the others. The content of T-AOC in the breasts of broilers fed 8 or 12 g/kg BFMA was higher than in the control group, and that of the broilers fed 8 g/kg BFMA was also higher than the group fed 4 g/kg. Relative to the control group, the lactic acid concentration in the breasts of broilers was significantly decreased by dietary supplementation with BFMA, and those in groups fed 8 or 12 g/kg BFMA were further decreased compared with the group fed 4 g/kg BFMA.

### 3.5. Cecum Microbiota

As shown in Figure 1, the α-diversity of cecal microbiota of broilers was indicated by Sobs, Shannon, Simpson, Ace, Chao, and coverage indexes of OTU level. Dietary supplementation with BFMA had no effect on the Shannon, Simpson, or coverage indexes of cecal microbiota of broilers on OTU level (*p* > 0.05). With increasing doses of BFMA addition in the diets of broilers, the Sobs, Ace, Chao, and coverage indexes of cecal microbiota showed significant decreases accordingly (*p* < 0.05). The β-diversity of cecal microbiota of broilers is presented in Figure 2. From the results of PLS-DA on OTU level point of view (Figure 2A), on the two-dimensional coordinate chart, each experimental group can be clustered together and the groups are obviously scattered (COMP1 = 8.72%, COMP2 = 6.52%). PCoA of weighted distance calculated from the phylum abundance matrix was performed to indicate the diversity of cecal microbiota of broilers on the phylum level (Figure 2B). The results (PC1 = 65.86%, PC2 = 10.65%) showed that each group was clustered together and all three groups treated with BFMA overlapped with the control group. However, there was no overlap between the group fed 12 g/kg malic acids in diet and the groups fed 4 or 8 g/kg BFMA, showing that there was a clear difference.

The microbial composition on genus level in cecum of the broilers is shown in Figure 3. The results of hierarchical clustering of the top 25 genera based on their abundance indicated that dietary supplementation with BFMA affected the genus composition of the cecal microbiota, relative to the control group (Figure 3A). The composition of cecal microbiota of broilers fed 8 and 12 g/kg BFMA in diets was most similar; then these two groups together clustered with the group fed the diet supplemented with 4 g/kg BFMA. The specific genus composition of cecal microbiota of broilers is presented in Figure 3B. All major bacterial genera were present in all groups, but in different proportions. The top six genera made up half of the total cecal microbiota, and they are in turn *Faecalibacterium*, *Bacteroides*, *Lactobacillus*, *unclassified_f_Lachnospiraceae*, *norank_f_norank_o_Clostridia_UCG-014*, and *Blautia*.

As shown in Figure 4, different species between dietary treatments at phylum and genus levels in the cecal microbiota of the broilers included two phyla, and eighteen genera (*p* < 0.05). Among the experimental groups, the different phyla were Campilobacterota and Verrucomicrobiota, and the different genera were *norank_f_norank_o_RF39*, *norank_f_Oscillospiraceae*, *Barnesiella*, *Helicobacter*, *Tuzzerella*, *Anaerostipes*, *Merdibacter*, *Caproiciproducens*, *Gordonibacter*, *Oscillospira*, *Roseburia*, *norank_f_norank_o_Rhodospirillales*, *Bifidobacterium, Victivallis*, *Achromobacter*, *Rhodococcus*, *unclassified_o_Bacteroidales*, and *ASF356*. Specifically, the proportions of *Campilobacterota* and *Verrucomicrobiota* in cecal microbiota of broilers fed BFMA were significantly decreased compared with the control group. The proportions of *norank_f_norank_o_RF39*, *norank_f_Oscillospiraceae*, *Helicobacter*, *Tuzzerella*, *Oscillospira*, *norank_f_norank_o_Rhodospirillales*, *Bifidobacterium*, *Victivallis*, and *ASF356* in cecal microbiota were significantly decreased by the dietary supplementation of BFMA. The proportions of *Barnesiella*, *Anaerostipes*, *Achromobacter*, *Rhodococcus*, and *unclassified_o_Bacteroidales* in cecal microbiota were increased by the addition of BFMA in the broilers’ diets. Relative to the control group, dietary supplementation with 8 g/kg BFMA significantly increased the amounts of *Merdibacter*, *Caproiciproducens*, and *Gordonibacter* in cecal microbiota of broilers, while these were decreased by the diets with the addition of 4 or 12 g/kg BFMA. The proportion of *Roseburia* in cecal microbiota was significantly increased by dietary supplementation with 4 g/kg BFMA, but was decreased with 8 or 12 g/kg BFMA.

## 4. Discussion

L-malic acid mainly acts as an antioxidant ingredient in the additives of the feed and food industries. Dietary supplementation of malic acid could reduce nitrogen emission pollution, and improve the performance and product quality of calves, cows, and lambs [28,29,30,31,32]. Although adding malic acid to the diet of broilers has no significant effect on growth performance [25], malic acid, an intermediate of the tricarboxylic acid (TCA) cycle, is hypothesized to influence the energy metabolism of muscle, whose effects on body health and meat quality of broilers are worthy of being disclosed. 

The L-malic acid product used in the present study was a mixture obtained from biological fermentation. The potential of organic acids as an in-feed antibiotic replacement to improve the intestinal health, feed efficiency and growth performance of broilers has been demonstrated [33,34,35]. It was confirmed again in this study that dietary supplementation with the organic acid, BFMA, improved the growth performance of broilers, including increasing ADFI, ADG, and BW, and decreasing FCR and DCR. Besides the benefits on growth performance, supplementation of mixed organic acids increased pH_24 h_ of the breast and thigh muscles, and the redness in thigh meat was also improved [36]. Organic acids prevent meat quality deterioration without leaving any chemical residues [37]. The combination of humic acid and organic acids improves the sensory attributes of cooked breast meat of broilers [38]. In the current study, dietary BFMA also increased the pH_24 h_ value and water-holding capacity of breast muscle of broilers without negative effects on carcass traits. Therefore, we conclude that BFMA, like other acidifiers, could improve the growth performance and meat quality of broilers.

Cells generate reactive oxygen species upon stress and accumulate the by-products of oxidation reactions, which could be eliminated by several antioxidant factors, mainly including GSH, SOD, and catalase [39]. The health status of broilers was improved by dietary supplementation with mixed organic acid through enhancing the immune function, the antioxidative characteristics and the tight junction proteins expression of the intestine [40]. Organic acids improved the welfare, health status, and cecal bacteria composition of broilers, but did not alter blood levels of antioxidant indices or liver function indicators [41]. MDA only slightly decreased with an increase in the level of formic plus lactic acid [42]. In the current study, dietary supplementation with BFMA significantly increased the serum immunity of broilers but did not influence serum antioxidant capacity. Antioxidants added in diets can minimize the negative effects of oxidative stress on the quality of broiler meat, and augment its water-holding and textural properties [43]. Judging from the indexes of GSH, GSH-Px, SOD, T-AOC, and IMP in the breast muscle of broilers, dietary supplementation with BFMA in the present study significantly enhanced the antioxidant capacity of breast muscle. Therefore, we deduced that BFMA probably increased the growth performance and body health of the broilers by reinforcing immunity, and decreased water loss of breast muscle via its antioxidant capacity against oxidative stress. Malic acid is one of the intermediates of the TCA cycle. Feeding tricarboxylic acid cycle intermediates improves lactate consumption in Chinese hamster ovary cell cultures [44]. In the current study, BFMA in the diets of broilers increased the pH_24 h_ value of breast muscle to a certain extent after slaughter at the age of 42 days. Therefore, it could be deducted that the decreased water loss of breast muscle of broilers was probably realized through reducing lactate accumulates and enhancing antioxidant capacity.

Intestinal microbiota could optimize the metabolic health of the host; how gut microbiota and their metabolites may link to the metabolism of the host or to the pathogenesis of metabolic disorders is a nascent and promising research field [45]. Besides performance, the chyme pH and microbiota of broilers were affected by acidifiers, 2-hydroxy-4-(methylthio) butanoic acid and DL-methionine [46]. The inclusion of benzoic acid in diets was effective in improving antioxidant capacity, nutrient digestion, and gut microbiota composition [17]. Dietary supplementation of a mixture of a free butyrate acidifier and gluconic acid improved gut health, including its morphological structure and microbiome activities, in chickens [47]. As an organic acid, addition of 2-hydroxy-4-methylthiobutyric acid in drinking water shows excellent bacteriostasis to benefit intestinal development and gut microbiota, and the subsequent performance of broilers [48]. Moreover, the tibia mass of broilers was improved by acidification of drinking water through the alterations of intestinal barrier and microbiota [16]. In the current study, the diversity of cecal microbiota of broilers was also significantly changed by dietary supplementation with BFMA, which was probably responsible for the enhanced immunity and performance.

The abundance of *o_RF39* in the gut was significantly decreased in rats with ulcerative colitis and severe colon tissue injury [49]. *Oscillospiraceae* in murine gut microbiota were found to be related to the development of rheumatoid arthritis and hyperlipidemia [50,51]. *Helicobacter* organisms are recognized as important pathogenic agents in colitic diseases of rodents and primates, potentially causing inflammatory bowel disease [52]. An abundance of *Tuzzerella* negatively affected the activity of methanogens [53]. The relative abundance of *Oscillospira* widely present in the animal and human intestines was positively associated with osteoporosis [54]. With the function of amino acid metabolism, *o_Rhodospirillales* is widely distributed in the environment with organic matter pollution. In the present study, dietary addition of BFMA significantly decreased the amount of *norank_f_norank_o_RF39*, *norank_f_Oscillospiraceae*, *Helicobacter*, *Tuzzerella*, *norank_f_norank_o_Rhodospirillales*, and *Oscillospira* in the cecum of broilers. This result indicated that the growth or reproduction of these harmful bacteria was effectively suppressed by BFMA.

*Barnesiella* intestinihominis could facilitate therapeutic immunomodulatory effects [55] and enable clearance of intestinal vancomycin-resistant Enterococcus colonization [56]. Conversion of dietary inositol into propionate and acetate by commensal *Anaerostipes* benefited host health [57]. *Merdibacter* was demonstrated to be a beneficial bacterium for gut health in laying hens [58]. The *Caproiciproducens* mainly degraded the proteins and carbohydrates from food residues to produce caproic acids through a chain elongation procedure [59]. The increase of *Gordonibacter* in gut microbiota of mice may reduce intestinal permeability and inflammatory cytokine levels, maintaining a healthy intestinal microenvironment [60]. *Roseburia*, a butyrate-producing genus, is considered a marker of gut health [61]. Members of the genus *Rhodococcus,* with versatile abilities to degrade various organic compounds, are able to generate triacylglycerols [62,63]. Intestinal Bacteroidales can play important roles in the bioactivities of sulfated polysaccharides [64]. In the current study, the abundances of *Barnesiella*, *Anaerostipes*, *Merdibacter*, *Caproiciproducens*, *Gordonibacter*, *Roseburia*, *Rhodococcus*, and *unclassified_o_Bacteroidales* in the cecum of broilers were significantly increased by the appropriate addition of BFMA in diets. Therefore, we deduced that these beneficial bacteria were enhanced to maintain the gut health of broilers.

*Bifidobacterium* has a diverse host range and shows several beneficial properties to the host [65]. *Victivallis*, a short-chain fatty acids (SCFAs) producer, increased significantly in irritable bowel syndrome patients with a starch- and sucrose-reduced dietary intervention [66]. Genus *ASF356*, also a SCFAs producing bacterium, increased along with the programming of hypertension prevention by maternal resveratrol supplementation [67] and decreased following Gastritis in mice [68]. *Achromobacter* was assumed to be an opportunistic pathogen [69]. In this study, dietary supplementation with BFMA decreased the abundance of *Bifidobacterium*, *Victivallis*, and *ASF356*, and increased that of *Achromobacter*. These adverse effects require attention when applying BFMA in broiler production.

## 5. Conclusions

As one of the intermediates of the TCA cycle, BFMA increases the water-holding capacity of breast muscle of broilers, probably through reducing lactate accumulates and enhancing antioxidant capacity. In addition, supplementation of BFMA in diets could improve performance at early stage and body health of broilers, probably by reinforcing immunity and perfecting the cecal microbiota structure. To be specific, harmful bacteria, including *norank_f_norank_o_RF39*, *norank_f_Oscillospiraceae*, *Helicobacter*, *Tuzzerella*, *norank_f_norank_o_Rhodospirillales*, and *Oscillospira,* were suppressed by BFMA. The probiotics, *Barnesiella*, *Anaerostipes*, *Merdibacter*, *Caproiciproducens*, *Gordonibacter*, *Roseburia*, *Rhodococcus*, and *unclassified_o_Bacteroidales*, were increased by BFMA. Meanwhile, it is alarming that BFMA decreased the abundance of probiotics *Bifidobacterium*, *Victivallis*, and *ASF356*, and increased the noxious bacterium *Achromobacter*.

## Figures and Tables

**Figure 1 antioxidants-11-02309-f001:**
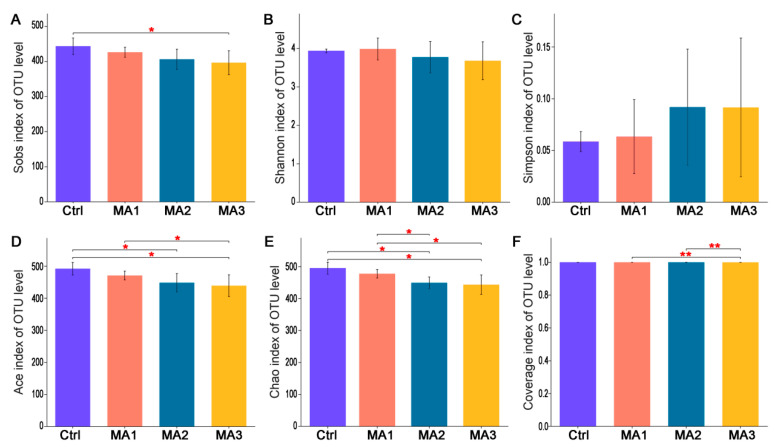
**The α**−**diversity of cecal microbiota of broilers.** It was indicated by Sobs (**A**), Shannon (**B**), Simpson (**C**), Ace (**D**), Chao (**E**), Coverage (**F**) indexes of operational taxonomic unit (OTU) level. Ctrl: the control group fed the basal diets; MA 1, 2, 3: the experimental groups fed the basal diets supplemented with 4, 8, and 12 g/kg of bio-fermented malic acid, respectively. * means significant differences between groups (*p* < 0.05). ** means significant differences between groups (*p* < 0.01).

**Figure 2 antioxidants-11-02309-f002:**
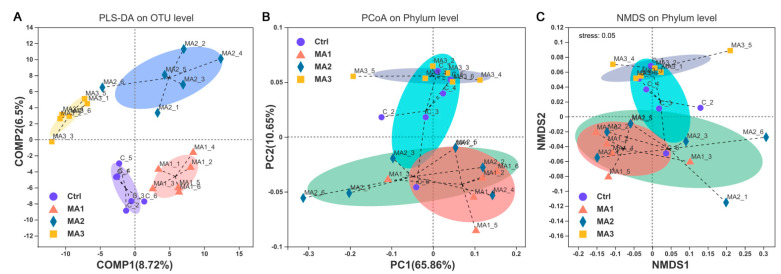
**The β**−**diversity of cecal microbiota of broilers.** (**A**) Partial least squares discriminant analysis (PLS-DA) on operational taxonomic unit (OTU) level. (**B**) Principal coordinate analysis (PCoA) of weighted distance calculated from phylum abundance matrix. (**C**) Nonmetric multidimensional scaling (NMDS) on phylum level. Ctrl: the control group; MA 1–3: the experimental groups fed the basal diets supplemented with 4, 8, and 12 g/kg of bio-fermented malic acid, respectively.

**Figure 3 antioxidants-11-02309-f003:**
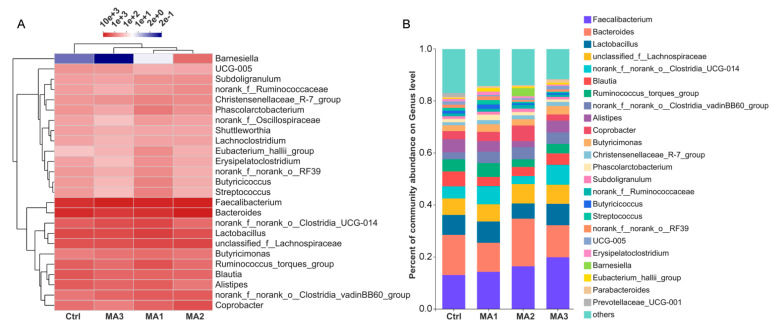
**The microbial composition on genus level in cecum of broilers.** (**A**) Hierarchical clustering of the top 25 genera based on their abundance. (**B**) The genus composition of cecal microbiota of broilers. Ctrl: the control group fed the basal diets; MA 1, 2, 3: the experimental groups fed the basal diets supplemented with 4, 8, and 12 g/kg of bio-fermented malic acid, respectively.

**Figure 4 antioxidants-11-02309-f004:**
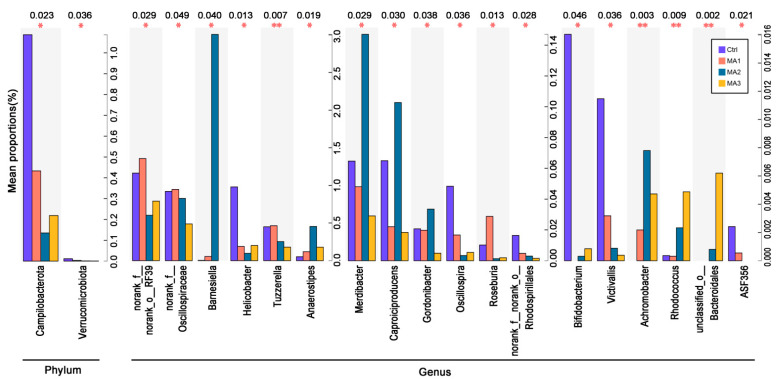
**Different species between dietary treatments at the phylum and genus level in the cecal microbiota of broilers.** Ctrl: the control group fed the basal diets; MA 1, 2, 3: the experimental groups fed the basal diets supplemented with 4, 8, and 12 g/kg of bio-fermented malic acid, respectively. * means significant differences between groups (*p* < 0.05). ** means significant differences between groups (*p* < 0.01).

**Table 1 antioxidants-11-02309-t001:** Ingredient and calculated nutrient compositions of basal diets.

Items	Days 1–14	Days 15–28	Days 29–42
Ingredients, %			
Corn	55.69	56.23	58.60
Soybean meal	35.33	33.48	31.26
Soybean oil	3.37	5.29	5.51
CaHPO_4_	2.27	2.00	1.82
Limestone	0.82	0.72	0.70
Salt	0.35	0.35	0.35
DL-Methionine	0.34	0.26	0.21
L-Lysine·HCl	0.31	0.15	0.03
Vitamin premix ^1^	0.02	0.02	0.02
Mineral premix ^2^	0.20	0.20	0.20
Choline chloride (50%)	0.10	0.10	0.10
Carrier of malic acid	1.20	1.20	1.20
Total	100.00	100.00	100.00
Nutrient levels ^3^			
AME, MJ/kg	12.35	12.97	13.18
Crude protein, %	22.00	21.00	20.00
Calcium, %	1.00	1.00	0.90
Available phosphorus, %	0.50	0.45	0.40
Lysine, %	1.29	1.15	1.09
Methionine, %	0.55	0.52	0.48
Methionine + cystine, %	0.94	0.84	0.84
Threonine, %	0.82	0.77	0.69
Tryptophan, %	0.24	0.18	0.22

^1^ Vitamin premix provided the following per kg of diets: VA, 12.500 IU; VD_3_, 2.500 IU; VK_3_, 2.65 mg; VB_1_, 2 mg; VB_2_, 6 mg; VB_12_, 0.025 mg; VE, 30 IU; biotin 0.0325 mg; folic acid, 1.25 mg; pantothenic acid, 12 mg; nicotinic acid, 50 mg. ^2^ Mineral premix provided the following per kg of diets: Cu 8 mg, Zn 75 mg, Fe 80 mg, Mn 100 mg, Se 0.15 mg, I 0.35 mg. ^3^ Nutrient levels were calculated values. AME, apparent metabolizable energy.

**Table 2 antioxidants-11-02309-t002:** Effects of dietary malic acid on the performance of broilers.

Item	BFMA, g/kg				SEM	*p*-Value
	0	4	8	12		
BW, g						
Day 1	49.70	49.70	49.70	49.70	0.00	1.000
Day 21	702.2 ^b^	860.56 ^a^	889.26 ^a^	866.39 ^a^	34.20	<0.001
Day 42	2585	2499	2646	2579	156.0	0.557
Day 1–21						
ADG, g	30.39 ^b^	39.01 ^a^	40.16 ^a^	39.28 ^a^	1.72	<0.001
ADFI, g	49.93 ^b^	53.80 ^a^	55.24 ^a^	55.53 ^a^	2.36	0.007
FCR	1.64 ^a^	1.38 ^b^	1.38 ^b^	1.41 ^b^	0.07	<0.001
DCR,%	0.11 ^a^	0.01 ^b^	0.01 ^b^	0.00 ^b^	0.03	<0.001
Day 22–42						
ADG, g	89.35	77.15	82.90	81.68	6.66	0.077
ADFI, g	147.53	139.22	149.24	145.17	11.00	0.521
FCR	1.66	1.81	1.80	1.78	0.09	0.111
DCR,%	0.00	0.07	0.02	0.03	0.04	0.272
Day 1–42						
ADG, g	56.49	56.81	60.26	59.52	3.96	0.369
ADFI, g	93.09	93.72	99.49	98.31	5.64	0.219
FCR	1.65	1.65	1.65	1.65	0.06	1.000
DCR, %	0.11	0.07	0.03	0.03	0.06	0.138

^a,b^ Means with no common superscripts differ significantly (*p* < 0.05). BFMA, bio-fermented malic acid; SEM, standard error of mean; BW, body weight; ADG, average daily gain; ADFI, average daily feed intake; FCR, feed conversion ratio (feed intake/weight gain, g:g); DCR, death and culling rate.

**Table 3 antioxidants-11-02309-t003:** Effects of dietary malic acid on the carcass traits and meat quality of broilers.

Item	BFMA, g/kg				SEM	*p*-Value
	0	4	8	12		
live BW, g	2604	2551	2749	2641	163.9	0.329
Carcass, g	2155	2118	2282	2196	150.0	0.414
Dressing, %	82.73	83.03	83.03	83.09	1.65	0.986
Eviscerated, g	1990	1945	2108	2015	135.4	0.339
Eviscerated, %	76.40	76.22	76.69	76.26	1.51	0.963
Breast, %	11.79	12.62	12.57	11.50	0.89	0.221
Thigh, %	14.84	14.90	14.16	15.27	1.00	0.464
Fat, %	1.93	2.34	2.26	2.19	0.30	0.228
Meat color						
*L* *	52.74	52.73	53.70	52.17	2.71	0.852
*a* *	6.33	6.93	6.61	6.11	1.81	0.913
*b* *	14.37 ^b^	19.70 ^a^	16.25 ^a,b^	14.13 ^b^	2.91	0.049
pH value						
45 min	6.55	6.38	6.48	6.44	0.21	0.692
24 h	5.40 ^b^	5.82 ^a^	5.74 ^a^	5.78 ^a^	0.19	0.033
Dropping loss, %	4.20 ^a^	3.44 ^b^	2.63 ^c^	3.28 ^b^	0.33	<0.001
Cooking loss, %	17.87 ^a^	17.52 ^a,b^	14.17 ^b^	16.83 ^a,b^	1.57	0.006

^a,b,c^ Means with no common superscripts differ significantly (*p* < 0.05). BFMA, bio-fermented malic acid; SEM, standard error of mean; BW, body weight; ADG, average daily gain; ADFI, average daily feed intake; FCR, feed conversion ratio (feed intake/weight gain, g:g); DCR, death and culling rate.

**Table 4 antioxidants-11-02309-t004:** Effects of dietary malic acid on serum antioxidant capacity of broilers.

Item	BFMA, g/kg				SEM	*p*-Value
	0	4	8	12		
Day 21						
GSH, mg/L	3.99	3.88	3.84	3.22	0.50	0.096
GSH-Px, 10^3^ U/mL	3.62	3.65	3.51	3.34	0.60	0.854
MDA, nmol/mL	3.94	3.90	3.42	3.63	0.69	0.614
SOD, U/mL	62.91 ^a^	64.95 ^a^	62.72 ^a^	50.38 ^b^	4.53	0.013
T-AOC, mM	1.17	1.31	1.38	1.56	0.39	0.546
Day 42						
GSH, mg/L	3.45	3.48	3.29	3.29	0.49	0.890
GSH-Px, 10^3^ U/mL	4.54	4.38	4.50	4.65	0.24	0.471
MDA, nmol/mL	4.96	5.37	5.19	4.78	0.92	0.781
SOD, U/mL	55.43	51.58	52.74	56.13	5.22	0.389
T-AOC, mM	1.42	1.57	1.59	1.61	0.26	0.709

^a,b^ Means with no common superscripts differ significantly (*p* < 0.05). BFMA, bio-fermented malic acid; SEM, standard error of mean; GSH, glutathione; GSH-Px, glutathione peroxidase; MDA, malondialdehyde; SOD, superoxide dismutase; T-AOC, total antioxidant capacity.

**Table 5 antioxidants-11-02309-t005:** Effects of dietary malic acid on serum immunity of broilers.

Item, g/L	BFMA, g/kg				SEM	*p*-Value
	0	4	8	12		
Day 21						
IgA	3.08 ^a^	2.84 ^a,b^	2.89 ^a,b^	2.77 ^b^	0.15	0.031
IgG	4.32 ^b^	4.26 ^b^	4.45 ^b^	4.66 ^a^	0.10	<0.001
IgM	1.77	1.83	2.05	2.07	0.20	0.075
Day 42						
IgA	2.75 ^b^	2.52 ^b^	3.36 ^a^	3.77 ^a^	0.27	<0.001
IgG	4.42 ^c^	4.41 ^b,c^	4.19 ^b^	4.48 ^a^	0.12	0.009
IgM	1.84	1.92	1.90	2.00	0.26	0.818

^a,b,c^ Mean with no common superscripts differ significantly (*p* < 0.05). BFMA, bio-fermented malic acid; SEM, standard error of mean; IgA, immune globulin A; IgG, immune globulin B; IgM, immune globulin M.

**Table 6 antioxidants-11-02309-t006:** Effects of dietary malic acid on antioxidant capacity of breast muscle of broilers.

Item	BFMA, g/kg				SEM	*p*-Value
	0	4	8	12		
GSH, 10^−2^ mg/g	21.73 ^b^	24.18 ^a,b^	27.52 ^a^	23.61 ^a,b^	3.44	0.039
GSH-Px, 10^3^ U/g	16.14 ^b^	19.82 ^a^	21.27 ^a^	20.15 ^a^	1.82	0.012
MDA, nmol/g	3.18	2.64	3.50	2.94	0.78	0.552
SOD, U/mg	82.26 ^b^	87.63 ^b^	122.48 ^a^	124.54 ^a^	13.95	0.008
T-AOC, μmol/g	5.69 ^c^	7.81 ^b,c^	11.98 ^a^	9.83 ^a,b^	2.79	0.032
IMP, mg/g	0.71 ^b^	1.05 ^a,b^	1.46 ^a^	1.34 ^a^	0.23	<0.001
Lactic acid, mol/g	0.43 ^a^	0.37 ^b^	0.31 ^c^	0.29 ^c^	0.03	0.026

^a,b,c^ Means with no common superscripts differ significantly (*p* < 0.05). BFMA, bio-fermented malic acid; SEM, standard error of mean; GSH, glutathione; GSH-Px, glutathione peroxidase; MDA, malondialdehyde; SOD, superoxide dismutase; T-AOC, total antioxidant capacity; IMP, inosine monophosphate.

## Data Availability

The raw data of this article will be available without reservation by contacting the corresponding author.

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
