# Peer review of "Bio-Fermented Malic Acid Facilitates the Production of High-Quality Chicken via Enhancing Muscle Antioxidant Capacity of Broilers"

_antioxidants, 2022, doi:10.3390/antiox11122309_

Round 1
Reviewer 1 Report (Previous Reviewer 1)
This manuscript was improved and appropriate for publication.
Author Response
Thank you.
Reviewer 2 Report (Previous Reviewer 2)
The Authors have clarified the various points and the work has certainly improved.
I advise the Authors to verify that the Title of the Tables and Figures are on the same page as the Table or Fig. To which they refer. (example: table 6 and Fig. 6)
Author Response
Thank you for your suggestion. We will pay attention to it when typesetting before publication.
Reviewer 3 Report (Previous Reviewer 3)
Thank you for your efforts
Authors did great efforts in improving the manuscript but I still not satisfied and have concern about the commercial product used and its composition,
The following link is for manuscript that puplished in antioxidant journal also. This manuscript concluded important impact for one of the ingredients (montmorillonite) that present in the current examined product
(https://doi.org/10.3390/antiox11091799). By the way, the examined material in this puplished manuscript represents 75 % of your examined product.... so I can't understand lines from 91 to 98... Please rewrite and add appropriate reference for this way. Please provide clear composition for the final used product. Otherwise you can modify your work to be examination for the whole product not mallic acid only.
Another concern is about the changes in some data in table 4, some of the data in table 4 changed completely, on what basis changed and where is the supplementery data before abd after changes
Author Response
Dear reviewer,
Thanks for your constructive comments. We have made detailed modifications carefully to improve our manuscript according to the comments and suggestions. The amendments made to the text are highlighted using the track changes mode in the revised manuscript. Our point-by-point responses are showed as follows. We do hope we could understand your questions correctly and have given right answers in the revised manuscript. Please feel free to inform me if there are still some questions. Thanks for your kind consideration.
Sincerely,
Shu-geng Wu, Ph.D. Professor
Institute of Feed Research, Chinese Academy of Agricultural Sciences, Beijing 100081, China.
Authors did great efforts in improving the manuscript but I still not satisfied and have concern about the commercial product used and its composition,
The following link is for manuscript that puplished in antioxidant journal also. This manuscript concluded important impact for one of the ingredients (montmorillonite) that present in the current examined product (https://doi.org/10.3390/antiox11091799). By the way, the examined material in this puplished manuscript represents 75 % of your examined product.... so I can't understand lines from 91 to 98... Please rewrite and add appropriate reference for this way. Please provide clear composition for the final used product. Otherwise you can modify your work to be examination for the whole product not mallic acid only.
Answer: I'm sorry to confuse the concepts of montmorillonite and zeolite powder. The carrier of malic acid in the product is mainly zeolite powder. The malic acid product has been evaluated in pig production, and we site the corresponding reference here. In the present study, the carrier content is same between experimental diets, and we have rewritten this part for easy understanding. Thank you for the suggestion.
Another concern is about the changes in some data in table 4, some of the data in table 4 changed completely, on what basis changed and where is the supplementary data before abd after changes
Answer: In Table 4, because the test values of MDA and SOD showed large within-group variations, we reviewed the raw data in time, and found that the large variation was caused by some abnormal test values, which was significantly greater or less than several times of the mean of the remaining values and obviously not within the normal range. Therefore, we conducted the chemical analysis again for relevant indicators and updated some of data correspondingly in Table 4. Thanks for your concern.
Round 2
Reviewer 3 Report (Previous Reviewer 3)
Authors did great efforts to improve their manuscript. No more comments. Thanks
This manuscript is a resubmission of an earlier submission. The following is a list of the peer review reports and author responses from that submission.
Round 1
Reviewer 1 Report
General comments:
Although this study indicates some interesting information, authors must provide the evidence that the beneficial effect of BFMA demonstrated in this paper is owing to the supplemental malic acid.
Specific comments:
1. Further study is necessary to investigate the effects of feeding L-malic acid alone, being equivalent to the dose of L-malic acid in BFMA ingested.
2. L90-91: Further information about the supplemental components such as montmorillonite, bacterial protein, polysaccharides, fermentation residue (in BFMA) in relation to gut microflora also should be explained. Especially, the effects of montmorillonite, bacterial protein, polysaccharides, and fermentation residue on gut microflora.
3. Although the results of lactate were indicated in Abstract (L33) and Conclusions (L 449), there were no data of lactate in this manuscript.
Author Response
Dear reviewer,
Thanks very much for your constructive comments and suggestions. We have improved the manuscript using the “Track Changes” mode of MS Word. Below you will find our point-by-point responses to comments. The whole manuscript has been carefully checked again by us. Please do not hesitate to contact us if you have any further questions.
Thanks for your time and efforts.
General comments:
Although this study indicates some interesting information, authors must provide the evidence that the beneficial effect of BFMA demonstrated in this paper is owing to the supplemental malic acid.
Answer: After consulting with the manufacturer of BFMA, we were told that the company not only produces BFMA (20% L-malic acid, 1% bacterial protein, 2% polysaccharide, 2% fermentation residue and 75% montmorillonite) for feed, but also produces pure malic acid extracted from BFMA. That is to say, the rest of BFMA after the extraction of malic acid was used to balance the addition of BFMA in experimental diets, rather than simple montmorillonite. So the other components except of malic acid are the same between the experimental groups. We are sorry for the mistake and have carefully modified the description in the main text and Table 1.
Specific comments:
- Further study is necessary to investigate the effects of feeding L-malic acid alone, being equivalent to the dose of L-malic acid in BFMA ingested.
Answer: Similar as the last question, the difference between treatment groups in the current experiment was only the dose of malic acid. The molecular structure and properties of L-malic acid produced by fermentation method were same with those produced by previous chemical synthesis, while the former has the advantages of low cost and no harmful substances, and can be directly used in food and feed without complex purification process.
- L90-91: Further information about the supplemental components such as montmorillonite, bacterial protein, polysaccharides, fermentation residue (in BFMA) in relation to gut microflora also should be explained. Especially, the effects of montmorillonite, bacterial protein, polysaccharides, and fermentation residue on gut microflora.
Answer: We agree with you that these components such as montmorillonite have an important impact on the gut microflora, but in the first question, we have explained that the content of these components, such as montmorillonite and other fermentation residues, are the same between the treatment groups in this study, only except for the amount of malic acid, which is in line with the experimental design to explore the effects of L-malic acid on broilers.
- Although the results of lactate were indicated in Abstract (L33) and Conclusions (L 449), there were no data of lactate in this manuscript.
Answer: Thank you for your constructive suggestion. The lactic acid content in chicken is indeed important for this study, so we supplemented the measurement of this index and added the data to Table 6. As expected, the concentration of lactic acid in breast meat was significantly decreased by the inclusion of BFMA in diets. We also supplemented corresponding contents in the sections of materials and results.
Reviewer 2 Report
The paper reports a study on the effects of antioxidant capacity of malic acid on the cecal microbial community and meat quality of broilers.
The Authors must provide information on the fermentation substrate and on any microorganisms used.
The main limitation of the study consists in the fact that the Authors did not consider the sex effect; and certainly at 42 d there is a dimorphism in the broiler.
Furthermore, if I have understood correctly from the M&M, all the results and analyzes are based on 6 replicates per treatment which for some parameters with high variability are certainly few.
Furthermore, I believe that due caution is required in the conclusions: 'Among the whole trial period, dietary supplementation of BFMA showed no effects on the ADFI, ADG, FCR, and DCR of broilers (P> 0.05)'. line 210
Author Response
Dear reviewer,
Thanks very much for your constructive comments and suggestions. We have improved the manuscript using the “Track Changes” mode of MS Word. Below you will find our point-by-point responses to comments. The whole manuscript has been carefully checked again by us. Please do not hesitate to contact us if you have any further questions.
Thanks for your time and efforts.
The paper reports a study on the effects of antioxidant capacity of malic acid on the cecal microbial community and meat quality of broilers.
The Authors must provide information on the fermentation substrate and on any microorganisms used.
Answer: Thank you for your suggestion. We supplemented the related information in the section of Material and methods as follows.
The bio-fermented malic acid (BFMA) was produced in Anhui Xuelang Biotechnology Co., Ltd (Bangbu, China) by the fermentation of aureobasidium pullulans CCTCC NO:M 2012223 with the substrate containing glucose, citric acid, and corn starch.
The main limitation of the study consists in the fact that the Authors did not consider the sex effect; and certainly at 42 d there is a dimorphism in the broiler.
Answer: We are sorry for the mistake. All of broilers used in the current study were male. We supplemented related information in the section of Material and methods.
Furthermore, if I have understood correctly from the M&M, all the results and analyzes are based on 6 replicates per treatment which for some parameters with high variability are certainly few.
Answer: The present test used 6 replicates per treatment with 12 chickens per replicate, which is sufficient and reliable for routine animal testing and majority of detection indicators, and is currently widely accepted and used. As you said, we did have some measures with large within-group variations, and other reviewer also raised this issue, such MDA and SOD in Table 4. We reviewed the raw data in time, and found that the large variation was caused by some abnormal test values. We have re-analyzed the relevant indicators to ensure the error within acceptable limits and updated some of data correspondingly in Table 4.
Furthermore, I believe that due caution is required in the conclusions: 'Among the whole trial period, dietary supplementation of BFMA showed no effects on the ADFI, ADG, FCR, and DCR of broilers (P> 0.05)'. line 210
Answer: Thank you for your suggestion. Dietary supplementation of BFMA only improved the performance of broilers at the early stage. We modified the related description in the conclusion according to your suggestion.
Reviewer 3 Report
The article entitled (Bio-fermented malic acid facilitates the production of high- 2 quality chicken via enhancing muscle antioxidant capacity of 3 broilers) contains valuable information and data, but needs further major concerns to be accepted in a prestigious journal as Antioxidants.
L 14: it is not scientific new born (plz write age of the broiler chicks)
L18-21: rewrite in scientific terms containing days.
L21-22: significant or not significant findings.
L41-42: correct to The antibiotic resistance and residues
L50: severe ????
L60: Salmonella Typhimurium (italic)
L62, 63: Campylobacter spp (italic)
L74-77: . In livestock production, feeding date palm leaves ensiled with malic acid in- 74 creased milk production, nutrient utilization efficiency, and milk quality [25] ,……………………….(delete). Give only information from citations concerning poultry
L89-91: The BFMA was produced by Anhui Xuelang Biotechnol- 89 ogy Co., Ltd (Bangbu, China), which contains 20% L-malic acid, 1% bacterial protein, 2% 90 polysaccharide, 2% fermentation residue and 75% montmorillonite……… here authors used commercial product contains many components, how can authors explains their results on the basis of Malic acid instead of others compounds effect. Authors here should use the purified extract of malic acid to direct the effects on their results according to Malic acid.
L91-93: The feeding trial 91 lasted for 42 days, and was divided into 3 stages: day 0-14 (starter period), day 15-28 92 (grower period), and day 29-42 (finisher period). If you have evidence that Arbor Acers guide lines proved three stages of growth please do this citations, most of broiler guidelines now have starter and finisher stages only.
L135: (L*, brightness; a*, Redness; b*, Yellowness), traits of colors should be italic and plz mention in details basis of giving your scores
At many places (after slaughter, give the age )
Please unify your abbreviations at many sections (ABW or BW)
Table4, SEM in SOD, U/mL, MDA, nmol/mL, SOD, U/mL is not acceptable and this means that there is sampling errors that leads to incorrect results and significance errors.
Conclusion section is conflicting and not accepted, one shot I felt that I am in another manuscript as referred in title
Author Response
Dear reviewer,
Thanks very much for your constructive comments and suggestions. We have improved the manuscript using the “Track Changes” mode of MS Word. Below you will find our point-by-point responses to comments. The whole manuscript has been carefully checked again by us. Please do not hesitate to contact us if you have any further questions.
Thanks for your time and efforts.
The article entitled (Bio-fermented malic acid facilitates the production of high- 2 quality chicken via enhancing muscle antioxidant capacity of 3 broilers) contains valuable information and data, but needs further major concerns to be accepted in a prestigious journal as Antioxidants.
L 14: it is not scientific new born (plz write age of the broiler chicks)
Answer: The age of birds is one-day. We modified the description the main text as follows. A total of 288 one-day-old Arbor acres (AA) male broiler chicks were randomly divided into 4 treatments.
L18-21: rewrite in scientific terms containing days.
Answer: We modified them according to your suggestion.
L21-22: significant or not significant findings.
Answer: We detailed the description.
L41-42: correct to The antibiotic resistance and residues
Answer: We modified it.
L50: severe ????
Answer: We changed it into “excessive”.
L60: Salmonella Typhimurium (italic)
Answer: We modified it.
L62, 63: Campylobacter spp (italic)
Answer: We modified them.
L74-77: . In livestock production, feeding date palm leaves ensiled with malic acid in- 74 creased milk production, nutrient utilization efficiency, and milk quality [25] ,……………………….(delete). Give only information from citations concerning poultry
Answer: We deleted it according to your suggestion.
L89-91: The BFMA was produced by Anhui Xuelang Biotechnol- 89 ogy Co., Ltd (Bangbu, China), which contains 20% L-malic acid, 1% bacterial protein, 2% 90 polysaccharide, 2% fermentation residue and 75% montmorillonite……… here authors used commercial product contains many components, how can authors explains their results on the basis of Malic acid instead of others compounds effect. Authors here should use the purified extract of malic acid to direct the effects on their results according to Malic acid.
Answer: After consulting with the manufacturer of BFMA, we were told that the company not only produces BFMA (20% L-malic acid, 1% bacterial protein, 2% polysaccharide, 2% fermentation residue and 75% montmorillonite) for feed, but also produces pure malic acid extracted from BFMA. That is to say, the rest of BFMA after the extraction of malic acid was used to balance the addition of BFMA in experimental diets, rather than simple montmorillonite. So the other components except of malic acid are the same between the experimental groups. We are sorry for the mistake and have carefully modified the description in the main text and Table 1.
L91-93: The feeding trial 91 lasted for 42 days, and was divided into 3 stages: day 0-14 (starter period), day 15-28 92 (grower period), and day 29-42 (finisher period). If you have evidence that Arbor Acers guide lines proved three stages of growth please do this citations, most of broiler guidelines now have starter and finisher stages only.
Answer: We agree with you that broiler feeding is currently mainly divided into two-stage or three-stage. The Arbor Acers Broiler Breeding Management Manual does not give out a strict restrictions on the divided stages of feeding. In China, due to the diversified market requirement of chicken, broilers are slaughtered at different age, so as a result, the three-stage feeding mode is widely accepted, which is more accurate for feeding than the two-stage type, and could increase economic benefits to a certain extent.
L135: (L*, brightness; a*, Redness; b*, Yellowness), traits of colors should be italic and plz mention in details basis of giving your scores
Answer: We modified them and supplemented the scoring details. The color of the meat is automatically detected by a colorimeter with a refer of whiteboard.
At many places (after slaughter, give the age )
Answer: We supplemented the age of birds.
Please unify your abbreviations at many sections (ABW or BW)
Answer: We changed ABW into BW.
Table4, SEM in SOD, U/mL, MDA, nmol/mL, SOD, U/mL is not acceptable and this means that there is sampling errors that leads to incorrect results and significance errors.
Answer: As you said, the within-group variations for MDA and SOD in Table 4 is too large. We reviewed the raw data in time, and found that the large variation was caused by some abnormal test values. We have re-analyzed the relevant indicators to ensure the error within acceptable limits and updated some of data correspondingly in Table 4.
Conclusion section is conflicting and not accepted, one shot I felt that I am in another manuscript as referred in title
Answer: Thank you for your suggestion. We adjusted the conclusion to make it more consistent with the title.
Round 2
Reviewer 1 Report
This revised manuscript still has fundamental problem, being important in the study.
Line 90-94: These sentences are strange and not clear because the specific extraction of only malic acid from BFMA is very difficult. Please explain the extraction method of only malic acid without extracting other components such as bacterial protein, polysaccharide, fermentation residues, and montmorllonite. Is it possible?
Style of references is incomplete and inconsistent.
Please check the page numbers of the following papers:
No. 17, 23, 24, 25, 28, 29, 31, 35, 41, 43, 47, 56, 63, 65, 67.
Reviewer 3 Report
Thank you for your response